# Application of micro-nanostructured magnetite in separating tetrabromobisphenol A and hexabromocyclododecane from environmental water by magnetic solid phase extraction

**Xiaoping Wang[1,2], Fengzhi He[1,2], Limin Zhang[1,2], Ang Yu[1,2]***

**1** Jiangsu Provincial Key Laboratory of Environmental Engineering, Jiangsu Provincial Academy of Environmental Science, Nanjing, Jiangsu, China, **2** Institute of Environmental Risk and Damage, Jiangsu Environmental Engineering Technology Co. LTD, Nanjing, Jiangsu, China

* yuangjsep@163.com

## Abstract

Two typical brominated flame retardants (BFRs), namely, tetrabromobisphenol A (TBBPA) and hexabromocyclododecane (HBCD), were persistent organic pollutants widely detected in various environmental media. This study aimed to successfully synthesize micro-nano-structured magnetite particles (MNMPs) with surface modification by citric acid molecules. The synthesized composites served as an adsorbent for extracting TBBPA and HBCD from environmental water samples followed by gas chromatography–mass spectrometry analysis. The obtained MNMPs were characterized in terms of crystal structure, morphology, size distribution, hydrophobic and hydrophilic performance and magnetism. The results indicated that the MNMPs exhibited high surface area, good dispersibility, and strong magnetic responsiveness for separation. The parameters affecting the extraction efficiency were optimized, including sample pH, amount of sorbents, extraction time and desorption conditions. Under the optimum conditions, the recovery was 83.5 and 107.1%, limit of detection was 0.13 and 0.35μg/mL ($S/N$ = 3), and limit of quantification was 0.37 and 0.59 μg/mL ($S/N$ = 10) for TBBPA and HBCD respectively. The relative standard deviations obtained using the proposed method were less than 8.7%, indicating that the MNMP magnetic solid-phase extraction method had advantages of simplicity, good sensitivity and high efficiency for the extraction of the two BFRs from environmental water.

## Introduction

Brominated flame retardants (BFRs) are adjuvants, that can reduce the combustion performance of flammable materials. Because BFRs contain the element bromine having good heat tolerance, excellent flame retardant performance, low price and other advantages, they are widely used in plastics, textiles, and electronic and electrical equipment [1]. More than 70

**Funding:** This research was funded by Jiangsu Provincial Key Laboratory of Environmental Engineering, which is a internal institution of Jiangsu Provincial Academy of Environmental Science. As a funder, it provided financial support, including experimental instruments platform service and research materials costs. The commercial company: Jiangsu Environmental Engineering Technology Co. LTD cooperated with this project research and paid authors' salaries.

**Competing interests:** All mentioned organizations neither served on the editorial board nor acted as an expert witness in relevant legal proceedings. Although the second author affiliations: Jiangsu Environmental Engineering Technology Co. LTD as a commercial affiliation, it does not alter our adherence to all PLOS ONE policies on sharing data and materials. It will not use the information to develop products, apply for patents.

varieties of BFRs exist with greater production and use, mainly including the polybrominated diphenylether, tetrabromobisphenol A (TBBPA), and hexabromocyclododecane (HBCD) [2]. However, BFRs have lipophilic properties. Hence, their mass production and application have led to serious ecological risks and physical and environmental hazards; they have become the persistent organic pollutants widely detected in various environmental media [3]. Therefore, studying the trend behavior and detection method of BFRs in typical environmental media is of great scientific significance.

The concentration of BFRs in various environmental media is found to be very low in trace analysis. Furthermore, isomerism makes the research on BFR detection and environmental behavior more difficult [4]. Many reports have been published in recent years on BFR analysis both at home and abroad. An internationally recognized analysis method for BFRs has not been published so far. Therefore, a simple and rapid analytical method, including sample pretreatment and detecting techniques, needs to be urgently developed, which can detect BFRs from environmental media accurately and efficiently. Currently, the predominantly adopted detection methods for BFRs are gas chromatography (GC) and liquid chromatography, combined with mass spectrometry (MS), considering the different volatility, polarity, and debromination for various BFRs [4–6]. Sample pretreatment plays an important role in the entire analysis process because it is related to not only the efficiency, but also the accuracy throughout the whole experimental analysis. The pretreatment techniques commonly used in the laboratory, are Soxhlet extraction, ultrasound extraction, microwave assisted extraction, and accelerated solvent extraction [4,7–9]. However, these methods have certain disadvantages, including long duration of extraction, consumption of large amounts of organic solvents, secondary pollution, and high cost.

Magnetite nanoparticles have recently been used as cost-effective adsorbents for the preconcentration of organic compounds in the environment samples due to their large surface area, superparamagnetism, and simple synthesis [10–14]. Due to high magnetization, they can be easily separated in an external magnetic field. More importantly, their properties, such as water dispersibility, electrostatic properties and stability, are controllable owing to various functional modifications, resulting in efficient and convenient enrichment for organic pollutants [15–18].

In this study, we synthesized micro-nanostructured magnetite particles (MNMPs) were synthesized using the hydrothermal method. The obtained composites were applied to extract BFRs from environmental water samples before gas chromatography–MS GC-MS analysis. This cost-effective method was successfully applied in the pretreatment of environmental water samples for BFR analysis. The real samples including tap and lake water were also analyzed by the proposed method.

## Materials and methods

### Standards and reagents

Ferrous chloride (FeCl$_2 \cdot$ 4H$_2$O) (purity $\geq$ 98%), anhydrous ferric chloride (FeCl$_3$) (purity $\geq$ 99%), sodium acetate anhydrous (NaOAc) (purity $\geq$ 99%), sodium chloride (NaCl) (purity $\geq$ 99%), sodium hydroxide(NaOH) (purity $\geq$ 99%), and trisodium citrate dihydrate (NaCit $\cdot$ 2H$_2$O) (purity $\geq$ 99%) were purchased from Shanghai Chemical Reagents Company, Shanghai, China. Ethylene glycol (EG), absolute ethyl alcohol, and hydrochloric acid (37%) were all analytical reagents purchased from Nanjing Chemical Reagents Company, Nanjing, China. TBBPA (purity$\geq$99%), HBCD (purity $\geq$ 99%), and acetic anhydride (Ac2O) (purity $\geq$ 99%) were purchased from Dr. Ehrensorfer Company, Hamburg, Germany, while GC grade methanol, acetone, ethyl acetate, acetonitrile, and hexane were obtained from

Merck, Darmstadt, Germany. Ultrapure water (18.2 MΩ) was obtained from a Milli-Q Gradient water system from Millipore, MA, USA.

**Instrumentation and analytical conditions.** The crystal structure of the synthesized materials was studied using x-ray diffraction (XRD) (Rigaku, Tokyo, Japan) with a graphite monochromator using CuKα radiation (λ = 1.5406 Å), within 20–80 2˚. The surface morphology was observed using a high-resolution JEM-2010 transmission electron microscope (TEM) (Japan Electron Optics Laboratory, Tokyo, Japan) and a SU-8010 scanning electron microscope (SEM) (Hitachi, Tokyo, Japan) with Electronic Data Switching line sweep for micro-area chemical analysis. The functional groups were investigated using a Nicolet iz10 Fourier transform infrared spectrometer (FTIR) (Thermo Scientific, MA, USA) in the range of 500–4000 $cm^{-1}$. Magnetic analysis was performed using an MPMS SQUID VSM-094 vibrating sample magnetometer (VSM) (Equistar Chemicals, NJ, USA). The specific surface areas of the materials were measured using an ASAP 2460 specific surface and porosity analyzer (Mack Instruments Inc, GA, USA). The size distribution was investigated using a Mastersizer 3000 Laser Diffraction Particle Size Analyzer (Malvern Instrument Co. Ltd., Malvern, UK). Wettability analyses were performed using the OCA20 contact angle measurement instruments (Dataphysics Instruments GmbH, Filderstadt, Germany).

Agilent GC-MS 6890N/5975B(Agilent Technologies Inc, CA, USA) was used to analyze the samples, which were separated using a DB-5MS capillary column (15 m×0.25 mm×0.1 μm). The initial temperature of the column oven was 100˚C, which was maintained for 1 min, raised to 280˚C with the rate of 15˚C/min and kept for 5 min. The source temperature and the quadrupole temperature were set at 230˚C and 150˚C, respectively. The carrier gas was highly pure helium with a flow rate of 1 mL/min. Electron impact ionization with a nominal electron energy of 70 eV was used. The SIM mode was selected for quantitative analysis. The retention time was 4 min. The ratio of mass-to-charge (*m/z*) of the characteristic ions for TBBPA and HBCD was 529 and 239 respectively [19].

## Synthesis of the magnetite adsorbent

The magnetite particles were synthesized by the hydrothermal treatment. First, $FeCl_3$(0.57 g) and $FeCl_2 \cdot 4H_2O$(0.30) were dissolved in EG (20 mL) to form a homogeneous solution with stirring. Second, NaOAc(1.20 g) as an alkali source and $NaCit \cdot 2H_2O$(0.20 g) as an electrostatic stabilizer were added to the reaction system and then completely dissolved with stirring. Third, the homogeneous mixture was added to a 50-mL stainless steel autoclave and heated at 200˚C for 10 h. It was cooled to room temperature, and the obtained black product was washed with ethanol and ultrapure water several times. Finally, it was dried in a vacuum oven at 60˚C for 6 h [20].

## Magnetic solid-phase extraction

The sample solution (20 mL) was placed in a 50-mL vitric tube. Next, 50 mg sorbent was directly added to it and vortexed for 20 min at room temperature. Subsequently, an external magnet on the side of the tube was used to collect the sorbent and remove the supernatant. Further, 2 mL of methanol was added to the tube to redissolve the analytes with stirring for 2 min vortex for three times. Then, 1 μL of the obtained eluent was injected for GC-MS analysis. The eluent was sequentially washed with 2 mL of acetonitrile under ultrasonication for 5 min to achieve sorbent regeneration and then dried before the next MSPE application. Fig 1 shows the description of the whole experimental process.

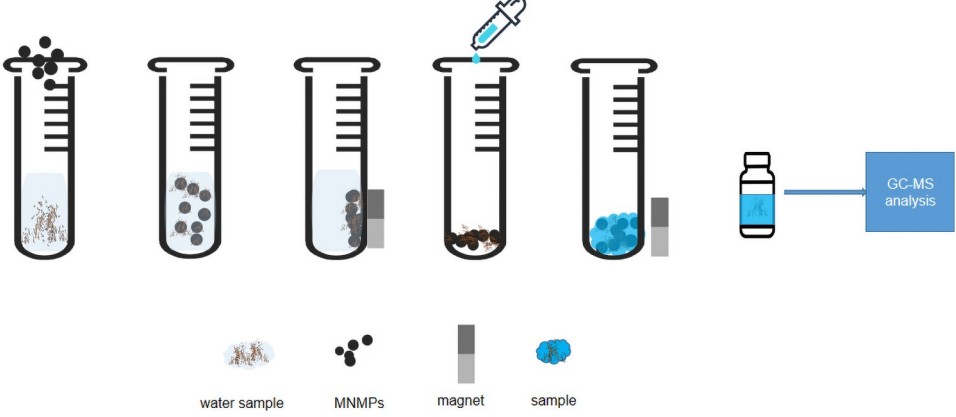

**Fig 1. Description of the whole experimental process.**

## Preparation of the water samples and standard solutions

The samples were prepared by dissolving the appropriate mother liquid of the analyte in 20 mL of ultrapure water. The individual 100 μg/mL stock standard solutions of TBBPA and HBCD were prepared in methanol and acetone, separately. A series of working standard solutions (1-–50 μg/mL) was prepared by dilute diluting the stock standard solutions to plot a matrix-matched calibration graph. The samples for TBBPA analysis were derived by using acetic anhydride before GC-MS analysis.

## Results and discussion

### Characterizations of MNMP adsorbent

**X-ray diffraction.** XRD patterns showed that the diffraction peaks surrounding at $2\theta$ = 30.2°, 35.3°, 43.7°, 53.9°, 57.1°, and 62.7° had good agreement with $Fe_3O_4$ (Reference code 01-089-3854); they belonged to the cubic structure system, corresponding to (220), (311), (400), (422), (511), and (440) facets of $Fe_3O_4$ [21]. Hence, the synthetic solids were intended $Fe_3O_4$ materials (Fig 2a).

**Morphological and elemental analysis.** The morphology and elemental composition of MNMPs were observed using an SEM with EDS and TEM. The SEM morphology of $Fe_3O_4$ is shown in Fig 2b. It had well-distributed loose clusters composed of nanocrystals with the size of about 5–10 nm; the particle size of agglomerate microspheres was approximately 250–400 nm. The presence of elements in $Fe_3O_4$ is shown in Fig 2c. The elements present in MNMPs were 31.15% O, 59.90% Fe and 8.95% C, suggesting thatMNMPs were successfully functionalized with NaCit · $2H_2O$. Fig 2d presents the typical TEM image. Under the transmission electron microscopic fields, the microspheres consisting of many magnetite nanocrystals were seen clearly. Hence, MNMPs had a large surface area for the adsorption of guest molecules, such as benzenoid compounds. The measured specific surface areas of synthetic $Fe_3O_4$ was about 81.25 $m^2$/g.

**FTIR spectra.** The surface groups of MNMPs were represented in the FTIR spectra (Fig 3a). The peak at 575.65 $cm^{-1}$ ascribed to the vibration of the Fe-O bond. The bands at 3383.01 $cm^{-1}$, 2968.88 $cm^{-1}$, 2916.32 $cm^{-1}$ and 1614.13 $cm^{-1}$ corresponded to O-H, C-H, C = O and C = C, respectively, which indicated the occurrence of carbonization under the surface modification with NaCit · $2H_2O$ [22]. The absorption peaks between 1000 $cm^{-1}$ to 1500 $cm^{-1}$ were

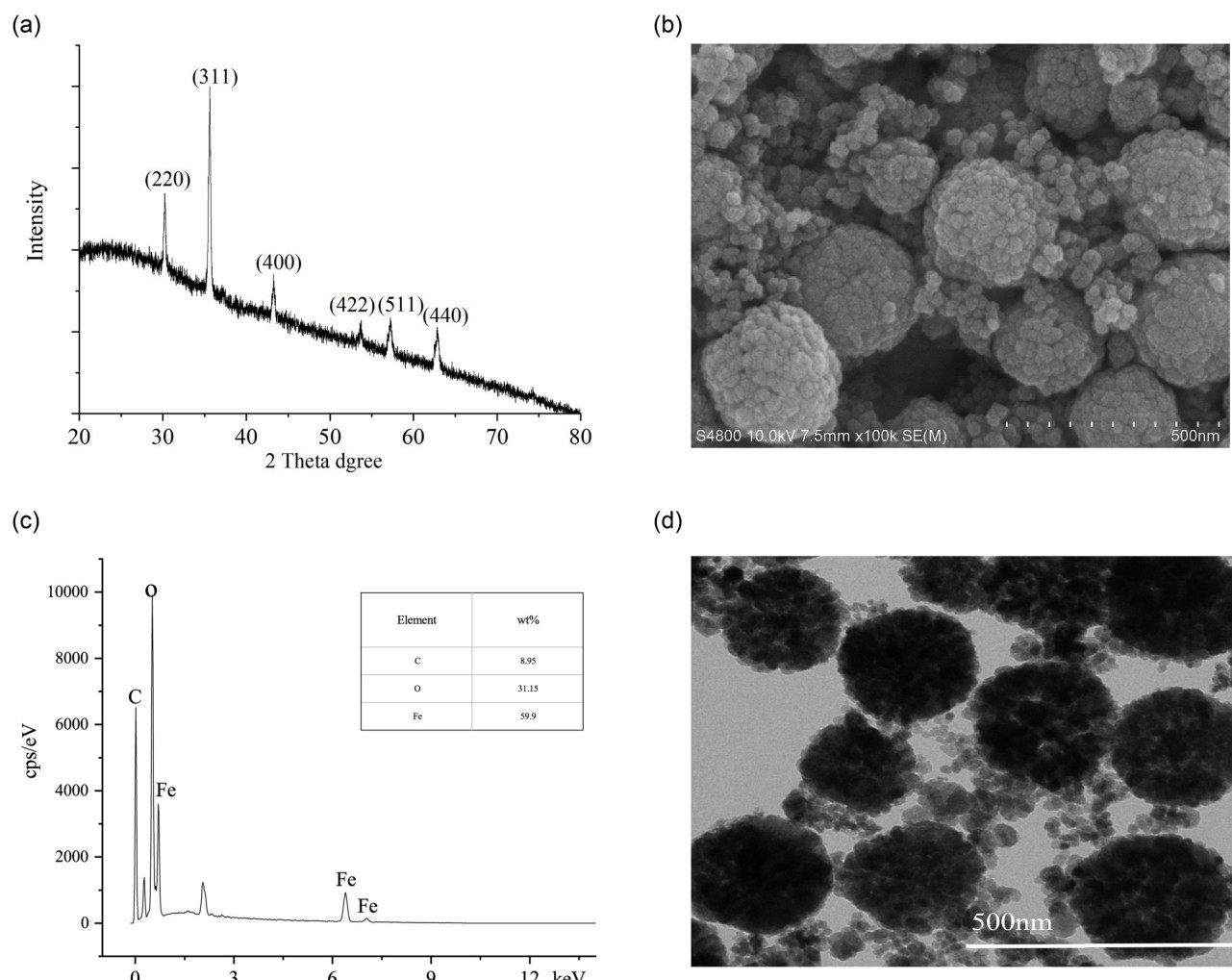

**Fig 2. Characterization of MNMP adsorbents.** (a) X-ray diffraction patterns. (b) Morphological analysis using SEM. (c) Distribution diagram of elements. (d) Image analysis using TEM.

associated with C-O stretching vibrations and O-H bending vibrations, suggesting a large number of hydrophilic groups on the surface of MNMPs [21]. The bands at 500 cm$^{-1}$ to 900 cm$^{-1}$ were assigned to the C-H out-of-plane bonding in benzene derivatives, which might facilitate the adsorption of benzenoid compounds using π-π interaction [21]. The aforementioned results reflected rich oxygen-containing groups on $Fe_3O_4$.

**Magnetic and dispersibility analyses.** The prepared MNMPs had a high saturation magnetization of 40.97 emu/g (Fig 3b), indicating superparamagnetic behavior at room temperature. The diameter distribution was is shown in Fig 3c; the average diameters of MNMPs was 350 nm. The wettability of MNMPs was investigated by the water contact angle (WCA) analysis. As shown in Fig 3d, a water droplet was found on MNMPs, which dispersed on the substrate, reflecting that the MNMPs were highly hydrophilic. The WCA measured was 21.90˚. Therefore, MNMPs could be easily dispersed in water and separated by an external magnetic field.

During the preparation process, EG acted as both the solvent and the reductant. Three carboxylate groups from NaCit · 2H$_2$O had a e strong coordination affinity to Fe(III) ions,

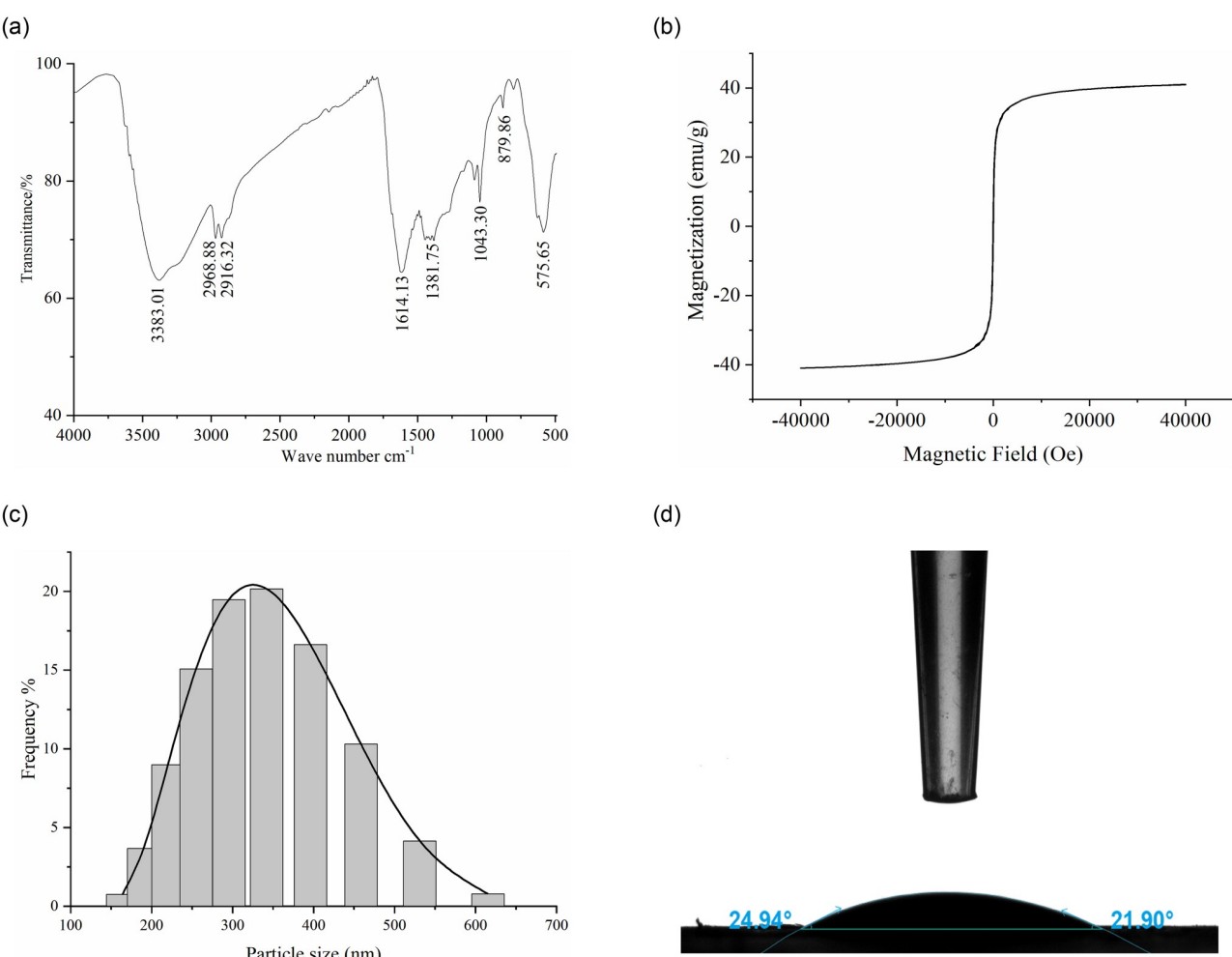

**Fig 3. Analysis of the physical and chemical properties of MNMPs.** (a) FTIR spectra analysis. (b) Hysteresis regression curve. (c) Diameter distribution. (d) Water contact angle analysis.

attached to the nanocrystals and prevented them from aggregating into large single crystals [20]. At the same time, NaCit · $2H_2O$ helped magnetite particles to form a polycrystalline structure, making them biocompatible and water dispersive; it also changed the charge density on the particle surface.

## Optimization of MSPE method

**Effect of sample pH.** The effect of sample solution pH on the recoveries was investigated in the range of 3.0–10.0 adjusted using HCl or NaOH solution. As shown in Fig 4a, high recoveries of the two selected BFRs were obtained at pH = 7.0. The recoveries of both were low at higher and lower pH. This phenomenon could be explained by combining the charged species and charge density on the surface of the sorbents [21]. The acidic conditions to the protonation of hydroxyl groups on the MNMP surface, which occupied the sorbent sites and made the sorbent more cationic. Incontras, alkaline conditions led to a large number of oxygen-groups on the adsorbent surface, which was ionized and adsorbed more water molecules. This hindered the adsorption of BFR molecules to the adsorption sites of MNMPs, leading to a decrease in extraction recoveries. TBBPA ($pKa_1 = 7.7 \pm 0.4$, $pKa_2 = 8.5 \pm 0.4$) had a hydroxyl

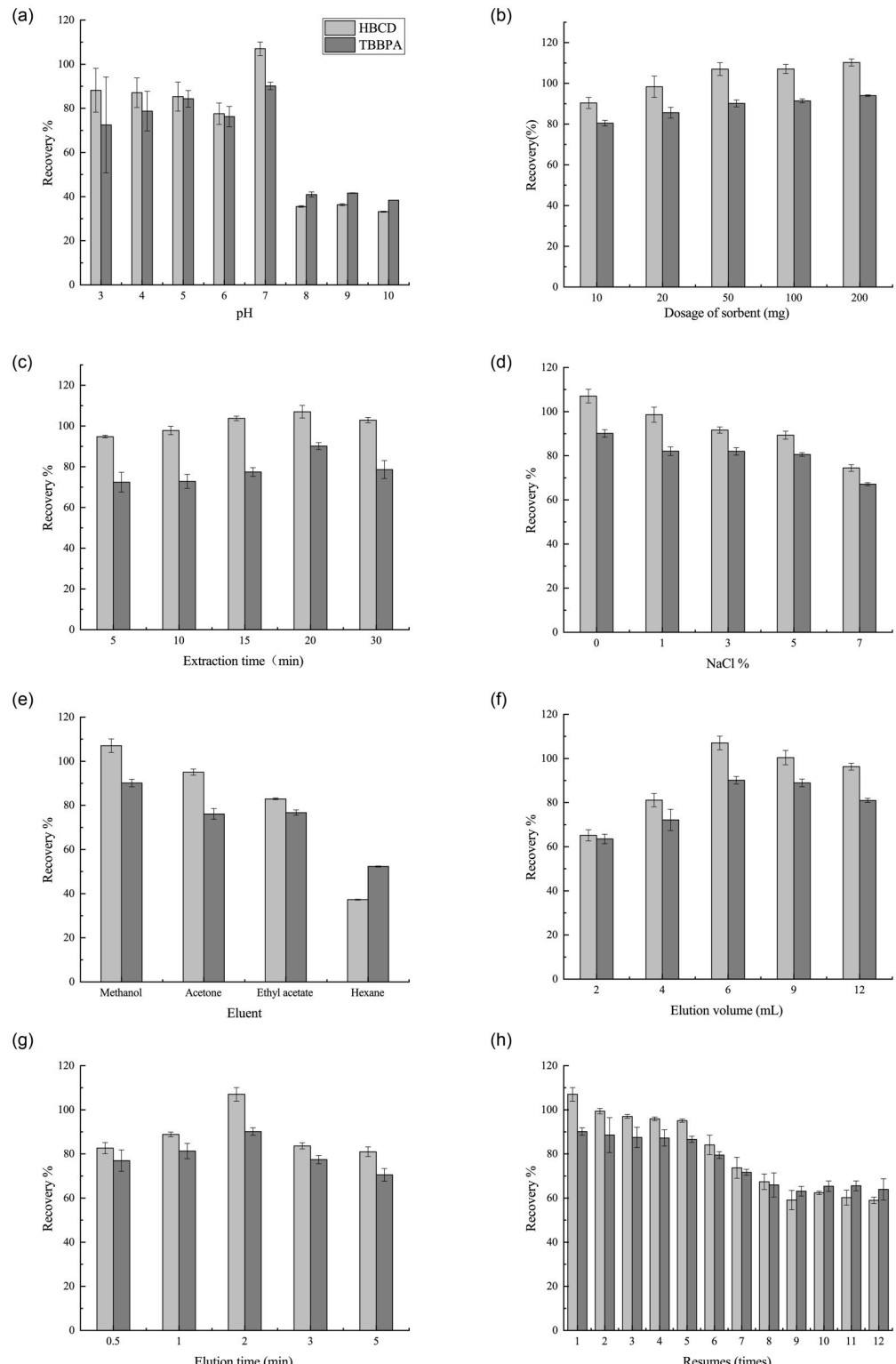

**Fig 4. Optimization of MSPE method.** (a) Effect of sample pH. (b) Effect of adsorbent dosage. (c) Effect of extraction time. (d) Effect of salt addition. (e) Effect of elution solvent. (f) Effect of elution volume (g) Effect of elution time. (h) Reusability of MNMPs. Spiked level: 20μg//mL.

group. In the acidic matrix, it could exist as a molecule. HBCD (Log$P$ = 6.63) was hydrophobic. The extraction efficiency decreased significantly for these ionizable analytes with the increase in pH, which attributed to the decreased hydrophobic interaction of MNMPs with dissociated TBBPA and HBCD. Moreover, $Fe_3O_4$ sorbents became $Fe(OH)_3$ under alkaline conditions, which intensified the decrease in the extraction recoveries of TBBPA and HBCD. Hence, neutral conditions facilitated the extraction process.

**Effect of adsorbent dosage.** The adsorbent dosage was chosen in the range of 10–200 mg to study the effect on recoveries (Fig 4b). The recoveries increased when the dosage of the adsorbent was increased from 10 to 50 mg, with no significant difference when more than 50 mg adsorbent was added. The results indicated that using 50 mg adsorbent to extract the TBBPA and HBCD from the sample solution was sufficient and effective. The adsorbent might have achieved its saturation uptake at the dosage of 50 mg [21,23]. Therefore, a further increase in dosage might have led to no significant difference in the extraction of analytes.

**Effect of extraction time.** A proper extraction time of 5–30 min was investigated in the present study. As shown in Fig 4c, a significant increase in sample recoveries was observed when the extraction time was increased from 5 to 20 min, indicating that abundant adsorption sites were present; the recoveries gradually increased with the extraction time. After 20 min, the recoveries decreased probably because of the desorption of analytes due to long vortex duration [23]. Therefore, 20 min was selected as the optimal extraction time.

**Effect of salt addition.** Different amounts of NaCl ranging from 0% to 7% (w/v) were added to investigate the effect of salt concentrations on the extraction recoveries of BFRs using MNMPs. As shown in Fig 4d, the extraction recoveries for BFRs decreased when the NaCl concentration increased. The results indicated that highly saline conditions negatively affected the solubility of analytes, which hindered the mass transfer of analytes from the solution to the adsorbent. Moreover, the adsorption sites of the MNMP surface might be occupied by $Na^+$ ions, thus intensifying the decrease in extraction efficiency [21]. As a result, no salt was added for further exploration.

**Effect of the elution solvent.** It is necessary to find out the elution solvent that can elute the target analytes from the MNMP adsorbents quickly and completely. Four different common organic solvents were chosen as eluents: methanol, acetone, hexane and ethyl acetate. Fig 4e shows the desorption capabilities of these eluents. The results indicated that methanol had a higher desorption capacity compared with other eluents. This fact could be explained by target analytes that were polar compounds and had high solubility in this solvent, which was in accordance with other findings [24]. Thus, methanol was selected as the desorption solvent.

**Effect of elution volume and time.** Additionally, the volume of desorption solvent ranging from 2 to 12 mL was chosen to investigate the influence of desorption efficiency (Fig 4f). The study found that 6 mL of methanol could desorb the analytes completely from the sorbent. Besides, the effect of desorption time between 0.5 and 5 min on the recoveries was also investigated. As shown in Fig 4g, a significant increment in recoveries was found when the desorption time increased from 0.5 to 2 min. At a desorption time of 2 min, the maximum recoveries were achieved, indicating that TBBPA and HBCD were completely desorbed from the adsorbent. When the elution time increased from 2 to 5 min, the obtained recoveries followed a downward trend. This phenomenon could be attributed to the loss of analytes and solvent evaporation [25]. Hence, the best desorption time selected was 2 min.

## Reusability of MNMPs

The reusability of MNMPs was investigated by washing the adsorbent twice with 2 mL of acetonitrile. and then drying it before use in the next MSPE. Fig 4h shows the recoveries of BFRs

Table 1. Analytical figures for MNMPs of TBBPA and HBCD.

| Analyte | Linearity (μg/mL) | $R^2$ | LOD (μg/mL) | LOQ (μg/mL) |
|---------|-------------------|-------|-------------|-------------|
| TBBPA | 1–50 | 0.996 | 0.13 | 0.37 |
| HBCD | 2–50 | 0.987 | 0.35 | 0.59 |

under multiple reuse. The results indicated that the recoveries of TBBPA and HBCD significantly decreased until the MNMP adsorbent was reused more than five times. Therefore, MNMPs had excellent properties of recycling in the sample pretreatment.

## Validation of the MNMP MSPE method

The performance of the MNMP MSPE method was investigated through evaluating various tested analytical parameters, including linearity, limit of detection (LOD), and limit of quantification (LOQ). As listed in Table 1, the linear calibration curves had a wide range, and the correlation of TBBPA and HBCD was 0.996 and 0.987, respectively. LOD and LOQ were defined as 3 times and 10 times the signal/slope of calibration curve, respectively. Combining the results achieved, the LOD value was found to be 0.13 and 0.35 μg/mL ($S/N = 3$), and the LOQ value was 0.37 and 0.59 μg/mL ($S/N = 10$) for TBBPA and HBCD respectively. The maximum permissible level (MRL) detected using this method for TBBPA and HBCD was 57.95 and 69.06 μg/mL, respectively. Therefore, the proposed MNMP MSPE method had good reliability.

## Analysis of real water samples

The developed method was used to determine TBBPA and HBCD in tap water and lake water samples, (marked as TW1, LW1, LW2). The water samples were spiked with the two analytes at the concentrations of 5, 10, and 20μg/mL to investigate the accuracy of the developed method, which were suitable concentrations for determining TBBPA and HBCD using performance peformence of the proposed MNMP MSPE pre-concentrate method. RSDs was calculated by applying the developed method for five repetition analyses, which included TBBPA and HBCD. The recoveries were in the range of 83.5%–107.1% with intraday RSDs < 8.7% and interday RSDs < 7.5% (Table 2). The acceptable recoveries indicated no effect of the matrix components in the real-water samples. The results implied that the developed method was reliable for determining BFRs in water samples.

## Comparisons with previously proposed methods

A comparison of the developed method with other previously reported preconcentration methods for TBBPA, HBCD, phenolic substances, and organic dyestuffs is presented in Table 3. The performances were evaluated in terms of determination method, LOD, linear range, and sample. The results showed that methods based on spectrophotometry and conventional preconcentration had problems in terms of selectivity and sensitivity. Compared with other proposed methods, the MNMP MSPE method had lower LOD and a good and useful linear range for TBBPA and HBCD, indicating that it was a sensitive method.

## Conclusions

In this study, the hydrothermal reaction was chosen to synthesize micro-nanostructured $Fe_3O_4$ composites. Then the obtained MNMPs were used as the extraction adsorbents for BFRs. After optimizing the extraction conditions, a rapid and sensitive MNMP MSPE-GC-MS

**Table 2. Precision and recoveries of TBBPA and HBCD in real environmental samples.**

| Sample | Spiked (μg/mL) | TBBPA (*n* = 5) | | | | HBCD (*n* = 5) | | | |
|---|---|---|---|---|---|---|---|---|---|
| | | Found (μg/mL) | Recovery (%) | RSD% | | Found μg/mL | Recovery (%) | RSD% | |
| | | | | Intraday | interday | | | Intraday | interday |
| TW1 | 0 | <LOD | – | – | – | <LOD | – | – | – |
| | 5 | 4.89±0.43 | 97.8 | 8.7 | 7.9 | 4.38±0.35 | 87.6 | 8.1 | 7.5 |
| | 10 | 9.48±0.36 | 94.8 | 3.8 | 6.7 | 9.14±0.52 | 91.4 | 5.7 | 3.8 |
| | 20 | 20.74±0.95 | 103.7 | 4.6 | 6.6 | 17.08±1.45 | 85.4 | 8.5 | 4.6 |
| LW1 | 0 | 0.48±0.04 | – | 7.3 | 5.8 | <LOD | – | – | – |
| | 5 | 5.84±0.48 | 107.1 | 8.2 | 4.4 | 4.51±0.13 | 90.1 | 2.8 | 4.6 |
| | 10 | 9.81±0.48 | 102.9 | 4.9 | 1.6 | 9.4±0.70 | 94.0 | 7.4 | 5.2 |
| | 20 | 20.04±1.67 | 97.8 | 8.3 | 2.3 | 17.52±1.42 | 87.6 | 8.1 | 2.9 |
| LW2 | 0 | <LOD | – | – | – | <LOD | – | – | – |
| | 5 | 4.74±0.23 | 94.8 | 4.8 | 5.7 | 4.57±0.17 | 91.4 | 3.7 | 3.5 |
| | 10 | 9.7±0.47 | 97.0 | 4.8 | 3.5 | 8.35±0.47 | 83.5 | 5.6 | 2.8 |
| | 20 | 18.7±1.22 | 93.5 | 6.5 | 2.8 | 18.02±0.43 | 90.1 | 2.4 | 7.4 |

**Table 3. Comparisons with other previously proposed methods.**

| Preconcentration method | Determination method | Target molecule | LOD | Linear range μg/mL | Applications | Ref. |
|---|---|---|---|---|---|---|
| Sequential injection | UV-vis | Paracetamol | 0.50 μg/mL | 0.5–75 | Pharmaceutical tablets | [26] |
| Liquid–liquid microextraction | HPLC-UV | Rhodamine | 2.90 μg/mL | Up to 100 | Industrial effluents | [27] |
| Solid-phase extraction | GC-MS | HBCD | 2.00 mg/kg | 1–50 | Coatings | [28] |
| Solid-phase extraction | GC-MS | HBCD | mg/kg | 1–50 | Environmental water | [29] |
| Liquid–liquid microextraction | GC-MS | HBCD | 30.00 mg/kg | 5–100 | Fireproof coatings | [30] |
| Liquid–liquid microextraction | GC-MS | TBBPA | 0.5 mg/kg | 1–50 | Coatings | [31] |
| Closed ultrasonic extraction | GC-MS | TBBPA | 50.00 mg/kg | 0.5–10 | Plastic | [32] |
| Magnetic solid-phase extraction | GC-MS | TBBPA | 0.13 μg/mL | 1–50 | Environmental water | This study |
| | | HBCD | 0.35 μg/mL | 2–50 | | |

method was established for determining of TBBPA and HBCD from environmental water. Moreover, the method was applied showed has shown high extraction efficiency.

## Supporting information

**S1 Data. Experimental datas for MSPE method optimization.**
(XLSX)

## Acknowledgments

I would like to acknowledge Experimental and Analytical Testing Center of Jiangsu Provincial Academy of Environmental Sciences, Nanjing, Jiangsu, China, for their sampling help from environmental water, which let the experiment go more smoothly.

## Author Contributions

**Conceptualization:** Ang Yu.

**Data curation:** Xiaoping Wang.

**Formal analysis:** Fengzhi He, Limin Zhang, Ang Yu.

**Funding acquisition:** Ang Yu.

**Investigation:** Xiaoping Wang, Fengzhi He, Limin Zhang, Ang Yu.

**Methodology:** Xiaoping Wang.

**Project administration:** Limin Zhang.

**Writing – original draft:** Fengzhi He.

**Writing – review & editing:** Xiaoping Wang, Fengzhi He, Limin Zhang, Ang Yu.

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
