## [Decision Letter · Decision Letter 0]

8 Feb 2021

PONE-D-20-40901

Application of micro-nanostructured magnetite in separating tetrabromobisphenol A and hexabromocyclododecane from environmental water by magnetic solid phase extraction

PLOS ONE

Dear Dr. yu,

Thank you for submitting your manuscript to PLOS ONE. After careful consideration, we feel that it has merit but does not fully meet PLOS ONE’s publication criteria as it currently stands. Therefore, we invite you to submit a revised version of the manuscript that addresses the points raised during the review process.

As appended below, the reviewers have raised major concern/critique and suggested further justification/work to consolidate the findings. Do go through the comments and amend the MS accordingly. Furthermore,

1- The MS should be edited by native speaker for grammar and syntax errors. At the moment, the language is not up to the standard

2- Line 34: Avoid using etc.,

3- Line 62-63: should not be here, delete. 

4- Full vendor details should include company, city (state), and country. Amend throughout the text

5- m/z and Hz should be italized throughout the MS

6- Method application should be applied to field incurred sample (not blank samples spiked with standards)

7- Comparison with other reported methods is needed to prove the feasibility of the developed method. This section should be before "Conclusions"

8- Inter-day accuracy and precision should also be performed (at least 3 days)

9- What are the reasons behind choosing these spiking levels (5, 10, 20)? Justification is needed. What are the MRL (maximum permissible level) of the tested analytes? This should be added and discussed in the text in relation to the sensitivity of the developed method

We look forward to receiving your revised manuscript.

Kind regards,

A. M. Abd El-Aty

Academic Editor

PLOS ONE

Journal Requirements:

3.Thank you for stating the following in the Financial Disclosure section:

"This research was funded by Jiangsu Provincial Key Laboratory of Environmental Engineering Opening Task ZX2018002"

We note that one or more of the authors are employed by a commercial company: Jiangsu Environmental Engineering Technology Co. LTD;

4. Please include your tables as part of your main manuscript and remove the individual files. Please note that supplementary tables (should remain/ be uploaded) as separate "supporting information" files.

Reviewers' comments:

Reviewer's Responses to Questions

**Comments to the Author**

1. Is the manuscript technically sound, and do the data support the conclusions?

Reviewer #1: Yes

Reviewer #2: Yes

Reviewer #3: Partly

2. Has the statistical analysis been performed appropriately and rigorously? 

Reviewer #1: Yes

Reviewer #2: Yes

Reviewer #3: N/A

3. Have the authors made all data underlying the findings in their manuscript fully available?

Reviewer #1: Yes

Reviewer #2: Yes

Reviewer #3: Yes

4. Is the manuscript presented in an intelligible fashion and written in standard English?

Reviewer #1: Yes

Reviewer #2: Yes

Reviewer #3: Yes

5. Review Comments to the Author

Reviewer #1: The paper "Application of micro-nanostructured magnetite in separating tetrabromobisphenol A and hexabromocyclododecane from environmental water by magnetic solid phase extraction" is a good paper.

All analytical figure of merit were reported and the whole project is clearly presented.

The new material is completely characterized.

The major point that require a deep revision is the validation procedure.

The Authors must follow the international guidelines in order to fully validate precision, trueness, LOD and LOQ.

These values must be validated considering the back-calculated concentrations and not merely the S/N ratios.

I suggest major revisions even if the paper merit the acceptance (but after these mandatory revisions)

Reviewer #2: In this study micro-nanostructured magnetite particles (MNMPs) were successfully synthesized with surface modification by citric acid molecules. Then the synthesized composites served as an adsorbent for the extraction of TBBPA and HBCD from environmental water samples followed by gas chromatography-mass spectrometry (GC-MS) analysis. The results indicated that MNMPs magnetic solid phase extraction (MSPE) method had advantages of simplicity, good sensitivity and high efficiency for the extraction of the two BFRs from environmental water. The manuscript is suitable for PLoS One.

Minor comments:

1. The established method only covers two BFRs, what about the applicability for the other BFRs?

2. Can this method be used for urine or blood samples in the future?

3. What is the meaning of “Environmental water”? River, lake or sea water? What about drinking water or tap water?

Reviewer #3: * Authors should check all significant digits throughout manuscript

* Effect of pH should be explained with pKa values of molecules in medium

* Is there any explanation from authors about desorption volume? Because, it is so high.(6 mL) for GC-MS analysis.

* How was LOD and LOQ values calculated? Which protocol was followed?

* All sections of figure 4 needs more comments? Why, How?

* In table 2, Recovery and RSD values should be corrected such as 102.9 instead of 102.886.

* These references may be added to introduction.

https://doi.org/10.1080/01496395.2019.1623254

https://doi.org/10.1016/j.microc.2018.11.056

https://doi.org/10.3390/molecules24244621

https://doi.org/10.1039/C9AY01504C

https://doi.org/10.1016/j.jchromb.2018.03.030

6. PLOS authors have the option to publish the peer review history of their article (what does this mean?). If published, this will include your full peer review and any attached files.

Reviewer #1: No

Reviewer #2: No

Reviewer #3: No

---

## [Author Response · Author response to Decision Letter 0]

6 Apr 2021

To Academic Editor：

Comment 1: The MS should be edited by native speaker for grammar and syntax errors. At the moment, the language is not up to the standard.

Answer: I'm sorry the manuscript turned out so many grammar and syntax errors. Very thanks for your suggestions. The language of the manuscript has been checked and revised with the help of a native speaker from professional language polishing institutions.

Comment 2: Line 34: Avoid using etc.

Answer: Thank you for your comments. It has been corrected in the revised manuscript.

Comment 3: Line 62-63: should not be here, delete. 

Answer: Thank you for your comments. It has been deleted in the revised manuscript.

Comment 4: Full vendor details should include company, city (state), and country. Amend throughout the text.

Answer: Thanks for your very thoughtful suggestion. We have added the full vendor details in the revised manuscript. See lines 64-90 for details. 

Comment 5: m/z and Hz should be italized throughout the MS.

Answer: Thanks for your very thoughtful suggestion. Do you mean m/z, S/N and n appeared in this manuscript should been italicized? If it is, we have revised them in our revised manuscript.

Comment 6: Method application should be applied to field incurred sample (not blank samples spiked with standards).

Answer: Thank you for your suggestion. We have applied this proposed method to analyse both TBBPA and HBCD from tap water and lake water samples. In order to investigate the accuracy and precision of the developed method, aforesaid water samples were also been spiked with standards for further analysis. Results were shown in Table 2.

Comment 7: Comparison with other reported methods is needed to prove the feasibility of the developed method. This section should be before "Conclusions".

Answer: Thanks for your very thoughtful suggestion. Comparisons between the developed method with other previous reported preconcentration methods was presented in Table 3. The performances were evaluated in terms of determination method, LOD, linear range and sample application. See more details in line 278-286, which was set before "Conclusions".

Comment 8: Inter-day accuracy and precision should also be performed (at least 3 days).

Answer: Thanks for your very thoughtful suggestion. Inter-day accuracy and precision (four days interval) have been supplied in Table 2. 

Comment 9: What are the reasons behind choosing these spiking levels (5, 10, 20)? Justification is needed. What are the MRL (maximum permissible level) of the tested analytes? This should be added and discussed in the text in relation to the sensitivity of the developed method.

Answer: Thanks for your very thoughtful suggestion. The reasons behind choosing these spiking levels (5, 10, 20) was expounded in line 265-267, which for the reason that “5, 10 and 20 μg/mL are suitable concentration level for determining TBBPA and HBCD using GC-MS instrument, helping to investigate the peformence of proposed MNMPs MSPE pre-concentrate method.” See from the references: 

Wang YY, Xu ZQ, Li D, Zheng JG，Zhou MH, Liu YF, et al. Determination of tetrabromobisphenol A in coatings by GC-MS method. J instrum anal. 2014;33(6):724-727. 

Zhang WJ, Hou XL, Han ZC. Determination of hexabromocyclododecane in water by gas chromatography mass spectrometry. Phy chem ins. 2017;53(9):1068-1070.

Tang ZK, Chen Q, Li D, Zhou MH, Liu YF, Qu CP, et al. Determination of hexabromocyclododecane in coatings by gas chromatography-mass spectrometry. Phy chem ins. 2012;48(7):807-811.

And the maximum permissible level (MRL) we detected from our experiment for TBBPA and HBCD was 57.95 and 69.06 μg/mL respectively. 

To Reviewer #1: 

Comment 1: The paper "Application of micro-nanostructured magnetite in separating tetrabromobisphenol A and hexabromocyclododecane from environmental water by magnetic solid phase extraction" is a good paper.

Answer: Many thanks for reviewer’s positive assessments and valuable comments that have enabled us to improve the manuscript.

Comment 2: All analytical figure of merit were reported and the whole project is clearly presented.

Answer: Many thanks for reviewer’s positive assessments and valuable comments that have enabled us to improve the manuscript.

Comment 3: The new material is completely characterized.

Answer: Many thanks for reviewer’s positive assessments.

Comment 4: The major point that require a deep revision is the validation procedure.

Answer: Many thanks for reviewer’s positive assessments and valuable comments that have enabled us to improve the manuscript. We have revised the validation procedure according to the following opinions. 

Comment 5: The Authors must follow the international guidelines in order to fully validate precision, trueness, LOD and LOQ. These values must be validated considering the back-calculated concentrations and not merely the S/N ratios.

Answer: Many thanks for reviewer’s valuable comments. We have checked the values of precision, trueness, LOD and LOQ in our manuscript. We have considering the effect from back signal noise.

Comment 6: I suggest major revisions even if the paper merit the acceptance (but after these mandatory revisions)

Answer: Many thanks for reviewer’s valuable comments. We have major revised this manuscript, especially the validation procedure. And the changes were marked in revised manuscript.

To Reviewer #2:

Comment 1: In this study micro-nanostructured magnetite particles (MNMPs) were successfully synthesized with surface modification by citric acid molecules. Then the synthesized composites served as an adsorbent for the extraction of TBBPA and HBCD from environmental water samples followed by gas chromatography-mass spectrometry (GC-MS) analysis. The results indicated that MNMPs magnetic solid phase extraction (MSPE) method had advantages of simplicity, good sensitivity and high efficiency for the extraction of the two BFRs from environmental water. The manuscript is suitable for PLoS One. Minor comments:

Answer: Many thanks for reviewer’s positive assessments and valuable comments that have enabled us to improve the manuscript. 

Comment 2: The established method only covers two BFRs, what about the applicability for the other BFRs?

Answer: Thank you for your comments. Tetrabromobisphenol A (TBBPA) and hexabromocyclododecane (HBCD) are the most wide used BFRs. It is of great significance to study the detection method of TBBPA and HBCD, which has been a research focus in the field of environmental contamination caused by BFRs. See from the references:

Yu G, Bu QW, Cao ZG, Du XM, Xia J, Wu M, et al. Brominated flame retardants (BFRs): A review on environmental contamination in China. Chemosphere. 2016; 150: 479-490.

Tang JF, Feng JY, Li XH, Li G. Levels of flame retardants HBCD, TBBPA and TBC in surface soils from an industrialized region of East China. Environ sci-Proc imp. 2014;16(5):1015-1021.

If this established method was applied for other BFRs, such as PBDEs, HBB and PBEB, it should us do further study to explore optimization conditions based on this research.

Comment 3: Can this method be used for urine or blood samples in the future?

Answer: Thank you for your comments. The proposed method was aimed to detect samples from environmental media, such as industrial effluents, fireproof coatings and environmental water. Urine or blood samples was biological samples, the preconcentration method parameters of them may turned out serious difference from environmental medias, should do further study in the future. 

Comment 4: What is the meaning of “Environmental water”? River, lake or sea water? What about drinking water or tap water?

Answer: I am sorry that I did not discuss the “environmental water ”clearly. The meaning of it was river, lake, tap water or industrial effluents. See from the references: 

Zhang WJ, Hou XL, Han ZC. Determination of hexabromocyclododecane in water by gas chromatography mass spectrometry. Phy chem ins. 2017;53(9):1068-1070

Biparva P, Ranjbari E, Hadjmohammadi MR. Application of dispersive liquid-liquid microextraction and spectrophotometric detection to the rapid determination of rhodamine 6G in industrial effluents. Anal chim acta. 2010;674(2):206-210.

However, due to the fact that sea water is salt containing, our proposed method was not apropriate for it. Besides, BFRs contained in drinking water with very low concentration level, our proposed method doesn't applicable for it either.

To Reviewer #3:

Comment 1: Authors should check all significant digits throughout manuscript.

Answer: Many thanks for reviewer’s valuable comments. We have check all significant digits throughout manuscript, especially the significant digits in validation section.

Comment 2: Effect of pH should be explained with pKa values of molecules in medium.

Answer: Many thanks for reviewer’s valuable comments. We have explained the effect of pH with pKa values of molecules in medium. See line 188-192 in details. Which help to comprehend the microcosmic mechanism.

Comment 3: Is there any explanation from authors about desorption volume? Because, it is so high.(6 mL) for GC-MS analysis.

Answer: Thank you for your comments. Desorption volume with 6mL was indeed so high for GC-MS analysis, but it was a fact that 6 mL methanol performed better to desorb the analytes from the sorbent in our experiments. Which showed similar results with the following research: 

Zhou QX, Lei M, Li J, Zhao KF, Liu YL. Determination of 1-naphthol and 2-naphthol from environmental waters by magnetic solid phase extraction with Fe@MgAl-layered double hydroxides nanoparticles as the adsorbents prior to high performance liquid chromatography. J chrom A. 2016; 1441:1-7. In their study, the volume of eluent was investigated in the range of 2-10mL. When the volume of eluent was 4 mL, it reached maximum peak area. 

Besides, Liu C, Ji YH, Jiang X, Yuan XC, Zhang XY, Zhao LS. The determination of pesticides in tea samples followed by magnetic multiwalled carbon nanotube-based magnetic solid-phase extraction and ultra-high performance liquid chromatography-tandem mass spectrometry. New j chem. In their study, different types (methanol, acetone, acetonitrile and ethyl acetate) and volumes (2, 4, 6, 8 and 10 ml) of desorption solvents were tested. The results showed that the recovery rate of the target compound reached the maximum when 6 ml of acetonitrile was used as the elution solvent. Which showed similar results with our study.

Comment 4: How was LOD and LOQ values calculated? Which protocol was followed?

Answer: Thank you for your comments. I am sorry that I did not discuss the LOD and LOQ clearly in our previous manuscript. The LOD and LOQ were defined as 3 times and 10 times the signal/slope of calibration curve, respectively.

Comment 5: All sections of figure 4 needs more comments? Why, How?

Answer: Thank you for your thoughtful comments. We have disccuss sections in figure 4 from the aspect of data trends, reason and mechanism. There still have some lackness, very thank you for more details comments on our next revison process.

Comment 6: In table 2, Recovery and RSD values should be corrected such as 102.9 instead of 102.886.

Answer: Thank you for your thoughtful comments. It has been corrected in the revised manuscript.

Comment 7: These references may be added to introduction.

https://doi.org/10.1080/01496395.2019.1623254

https://doi.org/10.1016/j.microc.2018.11.056

https://doi.org/10.3390/molecules24244621

https://doi.org/10.1039/C9AY01504C

https://doi.org/10.1016/j.jchromb.2018.03.030

Answer: Many thanks for reviewer’s references provision that have enabled us to improve the manuscript. We have added them into introduction. See line 50 and 54. Besides, these references indeed give us a deeper understanding of this congeneric research work.

---

## [Decision Letter · Decision Letter 1]

19 Apr 2021

Application of micro-nanostructured magnetite in separating tetrabromobisphenol A and hexabromocyclododecane from environmental water by magnetic solid phase extraction

PONE-D-20-40901R1

Dear Dr. yu,

We’re pleased to inform you that your manuscript has been judged scientifically suitable for publication and will be formally accepted for publication once it meets all outstanding technical requirements.

Kind regards,

A. M. Abd El-Aty

Academic Editor

PLOS ONE

Additional Editor Comments (optional):

Reviewers' comments:

Reviewer's Responses to Questions

**Comments to the Author**

1. If the authors have adequately addressed your comments raised in a previous round of review and you feel that this manuscript is now acceptable for publication, you may indicate that here to bypass the “Comments to the Author” section, enter your conflict of interest statement in the “Confidential to Editor” section, and submit your "Accept" recommendation.

Reviewer #1: All comments have been addressed

Reviewer #2: All comments have been addressed

Reviewer #3: All comments have been addressed

2. Is the manuscript technically sound, and do the data support the conclusions?

Reviewer #1: Yes

Reviewer #2: Yes

Reviewer #3: Yes

3. Has the statistical analysis been performed appropriately and rigorously? 

Reviewer #1: Yes

Reviewer #2: Yes

Reviewer #3: Yes

4. Have the authors made all data underlying the findings in their manuscript fully available?

Reviewer #1: Yes

Reviewer #2: Yes

Reviewer #3: Yes

5. Is the manuscript presented in an intelligible fashion and written in standard English?

Reviewer #1: Yes

Reviewer #2: Yes

Reviewer #3: Yes

6. Review Comments to the Author

Reviewer #1: All requested reviwions were reported in the current paper that, now, can be accepted for publication

Reviewer #2: (No Response)

Reviewer #3: (No Response)

7. PLOS authors have the option to publish the peer review history of their article (what does this mean?). If published, this will include your full peer review and any attached files.

Reviewer #1: No

Reviewer #2: No

Reviewer #3: **Yes: **Prof.Dr. Halil İbrahim Ulusoy

---

## [Editor Report · Acceptance letter]

23 Apr 2021

PONE-D-20-40901R1 

Application of micro-nanostructured magnetite in separating tetrabromobisphenol A and hexabromocyclododecane from environmental water by magnetic solid phase extraction 

Dear Dr. Yu:

I'm pleased to inform you that your manuscript has been deemed suitable for publication in PLOS ONE. Congratulations! Your manuscript is now with our production department. 

Kind regards, 

on behalf of

Prof. A. M. Abd El-Aty 

Academic Editor

PLOS ONE